# Partially-Supervised Image Captioning

**Peter Anderson**
Macquarie University*
Sydney, Australia
p.anderson@mq.edu.au

**Stephen Gould**
Australian National University
Canberra, Australia
stephen.gould@anu.edu.au

**Mark Johnson**
Macquarie University
Sydney, Australia
mark.johnson@mq.edu.au

## Abstract

Image captioning models are becoming increasingly successful at describing the content of images in restricted domains. However, if these models are to function in the wild — for example, as assistants for people with impaired vision — a much larger number and variety of visual concepts must be understood. To address this problem, we teach image captioning models new visual concepts from labeled images and object detection datasets. Since image labels and object classes can be interpreted as partial captions, we formulate this problem as learning from partially-specified sequence data. We then propose a novel algorithm for training sequence models, such as recurrent neural networks, on partially-specified sequences which we represent using finite state automata. In the context of image captioning, our method lifts the restriction that previously required image captioning models to be trained on paired image-sentence corpora only, or otherwise required specialized model architectures to take advantage of alternative data modalities. Applying our approach to an existing neural captioning model, we achieve state of the art results on the novel object captioning task using the COCO dataset. We further show that we can train a captioning model to describe new visual concepts from the Open Images dataset while maintaining competitive COCO evaluation scores.

## 1 Introduction

The task of automatically generating image descriptions, i.e., image captioning [1–3], is a long-standing and challenging problem in artificial intelligence that demands both visual and linguistic understanding. To be successful, captioning models must be able to identify and describe in natural language the most salient elements of an image, such as the objects present and their attributes, as well as the spatial and semantic relationships between objects [3]. The recent resurgence of interest in this task has been driven in part by the development of new and larger benchmark datasets such as Flickr 8K [4], Flickr 30K [5] and COCO Captions [6]. However, even the largest of these datasets, COCO Captions, is still based on a relatively small set of 91 underlying object classes. As a result, despite continual improvements to image captioning models and ever-improving COCO caption evaluation scores [7–10], captioning models trained on these datasets fail to generalize to images in the wild [11]. This limitation severely hinders the use of these models in real applications, for example as assistants for people with impaired vision [12].

In this work, we use weakly-annotated data (readily available in object detection datasets and labeled image datasets) to improve image captioning models by increasing the number and variety of visual concepts that can be successfully described. Compared to image captioning datasets such as COCO Captions, several existing object detection datasets [14] and labeled image datasets [15, 16] are much larger and contain many more visual concepts. For example, the recently released Open Images dataset V4 [14] contains 1.9M images human-annotated with object bounding boxes for 600 object

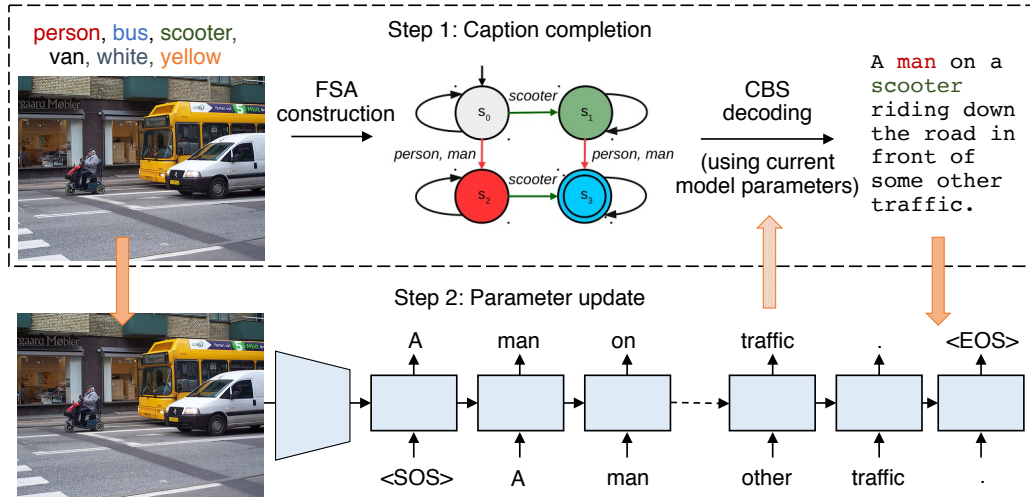

Figure 1: Conceptual overview of partially-specified sequence supervision (PS3) applied to image captioning. In Step 1 we construct finite state automata (FSA) to represent image captions partially-specified by object annotations, and use constrained beam search (CBS) decoding [13] to find high probability captions that are accepted by the FSA. In Step 2, we update the model parameters using the completed sequences as a training targets.

classes, compared to the 165K images and 91 underlying object classes in COCO Captions. This reflects the observation that, in general, object detection datasets may be easier to scale — possibly semi-automatically [17, 18] — to new concepts than image caption datasets. Therefore, in order to build more useful captioning models, finding ways to assimilate information from these other data modalities is of paramount importance.

To train image captioning models on object detections and labeled images, we formulate the problem as learning from partially-specified sequence data. For example, we might interpret an image labeled with 'scooter' as a partial caption containing the word 'scooter' and an unknown number of other missing words, which when combined with 'scooter' in the correct order constitute the complete sequence. If an image is annotated with the object class 'person', this may be interpreted to suggest that the complete caption description must mention 'person'. However, we may also wish to consider complete captions that reference the person using alternative words that are appropriate to specific image contexts — such as 'man', 'woman', 'cowboy' or 'biker'. Therefore, we characterize our uncertainty about the complete sequence by representing each partially-specified sequence as a finite state automaton (FSA) that encodes which sequences are consistent with the observed data. FSA are widely used in natural language processing because of their flexibility and expressiveness, and because there are well-known techniques for constructing and manipulating such automata (e.g., regular expressions can be compiled into FSA).

Given training data where the captions are either complete sequences or FSA representing partially-specified sequences, we propose a novel two-step algorithm inspired by expectation maximization (EM) [19, 20] to learn the parameters of a sequence model such as a recurrent neural network (RNN) which we will use to generate complete sequences at test time. As illustrated in Figure 1, in the first step we use constrained beam search decoding [13] to find high probability complete sequences that satisfy the FSA. In the second step, we learn or update the model parameters using the completed dataset. We dub this approach PS3, for partially-specified sequence supervision. In the context of image captioning, PS3 allows us to train captioning models jointly over both image caption and object detection datasets. Our method thus lifts the restriction that previously required image captioning models to be trained on paired image-sentence corpora only, or otherwise required specialized model architectures to be used in order to take advantage of other data modalities [21–24].

Consistent with previous work [13, 21–24], we evaluate our approach on the COCO novel object captioning splits in which all mentions of eight selected object classes have been eliminated from the caption training data. Applying PS3 to an existing open source neural captioning model [10], and

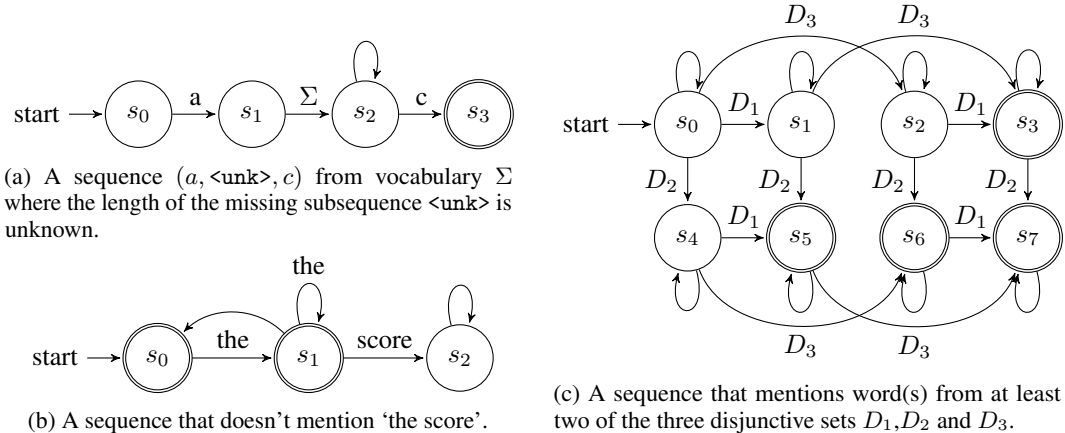

(a) A sequence $(a, \texttt{<unk>}, c)$ from vocabulary $\Sigma$ where the length of the missing subsequence $\texttt{<unk>}$ is unknown.

(b) A sequence that doesn't mention 'the score'.

(c) A sequence that mentions word(s) from at least two of the three disjunctive sets $D_1$, $D_2$ and $D_3$.

Figure 2: PS3 is a general approach to training RNNs on partially-specified sequences. Here we illustrate some examples of partially-specified sequences that can be represented with finite state automata. Unlabeled edges indicate 'default transitions', i.e., an unlabeled edge leaving a node $n$ is implicitly labeled with $\Sigma \setminus S$, where $S$ is the set of symbols on labeled edges leaving $n$ and $\Sigma$ is the complete vocabulary.

training on auxiliary data consisting of either image labels or object annotations, we achieve state of the art results on this task. Furthermore, we conduct experiments training on the Open Images dataset, demonstrating that using our method a captioning model can be trained to identify new visual concepts from the Open Images dataset while maintaining competitive COCO evaluation scores.

Our main contributions are threefold. First, we propose PS3, a novel algorithm for training sequence models such as RNNs on partially-specified sequences represented by FSA (which includes sequences with missing words as a special case). Second, we apply our approach to the problem of training image captioning models from object detection and labeled image datasets, enabling arbitrary image captioning models to be trained on these datasets for the first time. Third, we achieve state of the art results for novel object captioning, and further demonstrate the application of our approach to the Open Images dataset. To encourage future work, we have released our code and trained models via the project website[2]. As illustrated by the examples in Figure 2, PS3 is a general approach to training sequence models that may be applicable to various other problem domains with partially-specified training sequences.

## 2 Related work

**Image captioning**   The problem of image captioning has been intensively studied. More recent approaches typically combine a pretrained Convolutional Neural Network (CNN) image encoder with a Recurrent Neural Network (RNN) decoder that is trained to predict the next output word, conditioned on the previous output words and the image [1, 25–28], optionally using visual attention [2, 7–10]. Like other sequence-based neural networks [29–32], these models are typically decoded by searching over output sequences either greedily or using beam search. As outlined in Section 3, our proposed partially-supervised training algorithm is applicable to this entire class of sequence models.

**Novel object captioning**   A number of previous works have studied the problem of captioning images containing novel objects (i.e., objects not present in training captions) by learning from image labels. Many of the proposed approaches have been architectural in nature. The Deep Compositional Captioner (DCC) [21] and the Novel Object Captioner (NOC) [22] both decompose the captioning model into separate visual and textual pipelines. The visual pipeline consists of a CNN image classifier that is trained to predict words that are relevant to an image, including the novel objects. The textual pipeline is a RNN trained on language data to estimate probabilities over word sequences. Each pipeline is pre-trained separately, then fine-tuned jointly using the available image and caption data. More recently, approaches based on constrained beam search [13], word copying [33] and

neural slot-filling [24] have been proposed to incorporate novel word predictions from an image classifier into the output of a captioning model. In contrast to the specialized architectures previously proposed for handling novel objects [21–24], we present a general approach to training sequence models on partially-specified data that uses constrained beam search [13] as a subroutine.

**Sequence learning with partial supervision**    Many previous works on semi-supervised sequence learning focus on using *unlabeled sequence data* to improve learning, for example by pre-training RNNs [34, 35] or word embeddings [36, 37] on large text corpora. Instead, we focus on the scenario in which the sequences are *incomplete or only partially-specified*, which occurs in many practical applications ranging from speech recognition [38] to healthcare [39]. To the best of our knowledge we are the first to consider using finite state automata as a new way of representing labels that strictly generalizes both complete and partially-specified sequences.

# 3    Partially-specified sequence supervision (PS3)

In this section, we describe how partially-specified data can be incorporated into the training of a sequence prediction model. We assume a model parameterized by $\theta$ that represents the distribution over complete output sequences $\boldsymbol{y} = (y_1, \ldots, y_T), \boldsymbol{y} \in Y$ as a product of conditional distributions:

$$p_\theta(\boldsymbol{y}) = \prod_{t=1}^{T} p_\theta(y_t \mid y_{1:t-1}) \tag{1}$$

where each $y_t$ is a word or other token from vocabulary $\Sigma$. This model family includes recurrent neural networks (RNNs) and auto-regressive convolutional neural networks (CNNs) [29] with application to tasks such as language modeling [30], machine translation [31, 32], and image captioning [1–3]. We further assume that we have a dataset of partially-specified training sequences $X = \{\boldsymbol{x}^0, \ldots, \boldsymbol{x}^m\}$, and we propose an algorithm that simultaneously estimates the parameters of the model $\theta$ and the complete sequence data $Y$.

## 3.1    Finite state automaton specification for partial sequences

Traditionally partially-specified data $X$ is characterized as incomplete data containing missing values [19, 40], i.e., some sequence elements are replaced by an unknown word symbol <unk>. However, this formulation is insufficiently flexible for our application, so we propose a more general representation that encompasses missing values as a special case. We represent each partially-specified sequence $\boldsymbol{x}^i \in X$ with a finite state automaton (FSA) $A^i$ that *recognizes* sequences that are consistent with the observed partial information. Formally, $A^i = (\Sigma, S^i, s_0^i, \delta^i, F^i)$ where $\Sigma$ is the model vocabulary, $S^i$ is the set of automaton states, $s_0^i \in S^i$ is the initial state, $\delta^i : S^i \times \Sigma \to S^i$ is the state-transition function that maps states and words to states, and $F^i \subseteq S^i$ is the set of final or accepting states [41].

As illustrated in Figure 2, this approach can encode very expressive uncertainties about the partially-specified sequence. For example, we can allow for missing subsequences of unknown or bounded length, negative information, and observed constraints in the form of conjunctions of disjunctions or partial orderings. Given this flexibility, from a modeling perspective the key challenge in implementing the proposed approach will be determining the appropriate FSA to encode the observed partial information. We discuss this further from the perspective of image captioning in Section 4.

## 3.2    Training algorithm

We now present a high level specification of the proposed PS3 training algorithm. Given a dataset of partially-specified training sequences $X$ and current model parameters $\theta$, then iteratively perform the following two steps:

Step 1.  Estimate the complete data $Y$ by setting $\boldsymbol{y}^i \leftarrow \operatorname{argmax}_{\boldsymbol{y}} p_\theta(\boldsymbol{y} \mid A^i)$ for all $\boldsymbol{x}^i \in X$
Step 2.  Learn the model parameters by setting $\theta \leftarrow \operatorname{argmax}_\theta \sum_{\boldsymbol{y} \in Y} \log p_\theta(\boldsymbol{y})$

Step 1 can be skipped for complete sequences, but for partially-specified sequences Step 1 requires us to find the most likely output sequence that satisfies the constraints specified by an FSA. As it is typically computationally infeasible to solve this problem exactly, we use *constrained beam*

---

**Algorithm 1** Beam search decoding

---

1: **procedure** BS($\Theta, b, T, \Sigma$)  $\quad\quad\quad\quad\quad\quad\quad\triangleright$ With beam size $b$ and vocabulary $\Sigma$
2: $\quad B \leftarrow \{\epsilon\}$  $\quad\quad\quad\quad\quad\quad\quad\quad\quad\quad\quad\quad\quad\quad\quad\triangleright \epsilon$ is the empty string
3: $\quad$ **for** $t = 1, \ldots, T$ **do**
4: $\quad\quad E \leftarrow \{(\boldsymbol{y}, w) \mid \boldsymbol{y} \in B, w \in \Sigma\}$  $\quad\quad\triangleright$ All one-word extensions of sequences in $B$
5: $\quad\quad B \leftarrow \mathrm{argmax}_{E' \subset E, |E'|=b} \sum_{\boldsymbol{y} \in E'} \Theta(\boldsymbol{y})$  $\quad\triangleright$ The $b$ most probable extensions in $E$
6: $\quad$ **return** $\mathrm{argmax}_{\boldsymbol{y} \in B} \Theta(\boldsymbol{y})$  $\quad\quad\quad\quad\quad\quad\quad\triangleright$ The most probable sequence

---

---

**Algorithm 2** Constrained beam search decoding [13]

---

1: **procedure** CBS($\Theta, b, T, A = (\Sigma, S, s_0, \delta, F)$)  $\quad\quad\quad\quad\triangleright$ With finite state recognizer $A$
2: $\quad$ **for** $s \in S$ **do**
3: $\quad\quad B^s \leftarrow \{\epsilon\}$ if $s = s_0$ else $\emptyset$  $\quad\quad\quad\quad\quad\quad\triangleright$ Each state $s$ has a beam $B^s$
4: $\quad$ **for** $t = 1, \ldots, T$ **do**
5: $\quad\quad$ **for** $s \in S$ **do**  $\quad\quad\quad\quad\triangleright$ Extend sequences through state-transition function $\delta$
6: $\quad\quad\quad E^s \leftarrow \cup_{s' \in S} \{(\boldsymbol{y}, w) \mid \boldsymbol{y} \in B^{s'}, w \in \Sigma, \delta(s', w) = s\}$
7: $\quad\quad\quad B^s \leftarrow \mathrm{argmax}_{E' \subset E^s, |E'|=b} \sum_{\boldsymbol{y} \in E'} \Theta(\boldsymbol{y})$  $\quad\triangleright$ The $b$ most probable extensions in $E^s$
8: $\quad$ **return** $\mathrm{argmax}_{\boldsymbol{y} \in \bigcup_{s \in F} B^s} \Theta(\boldsymbol{y})$  $\quad\quad\quad\triangleright$ The most probable accepted sequence

---

*search* [13] to find an approximate solution. In Algorithms 1 and 2 we provide an overview of the constrained beam search algorithm, contrasting it with beam search [42]. Both algorithms take as inputs a scoring function which we define by $\Theta(\boldsymbol{y}) = \log p_\theta(\boldsymbol{y})$, a beam size $b$, the maximum sequence length $T$ and the model vocabulary $\Sigma$. However, the constrained beam search algorithm additionally takes a finite state recognizer $A$ as input, and guarantees that the sequence returned will be accepted by the recognizer. Refer to Anderson et al. [13] for a more complete description of constrained beam search. We also note that other variations of constrained beam search decoding have been proposed [43–45]; we leave it to future work to determine if they could be used here.

**Online version** The PS3 training algorithm, as presented so far, is inherently offline. It requires multiple iterations through the training data, which can become impractical with large models and datasets. However, our approach can be adapted to an online implementation. For example, when training neural networks, Steps 1 and 2 can be performed for each minibatch, such that Step 1 estimates the complete data for the current minibatch $Y' \subset Y$, and Step 2 performs a gradient update based on $Y'$. In terms of implementation, Steps 1 and 2 can be implemented in separate networks with tied weights, or in a single network by backpropagating through the resulting search tree in the manner of Wiseman and Rush [46]. In our GPU-based implementation, we use separate networks with tied weights. This is more memory efficient when the number of beams $b$ and the number of states $|S|$ is large, because performing the backward pass in the smaller Step 2 network means that it is not necessary to maintain the full unrolled history of the search tree in memory.

**Computational complexity** Compared to training on complete sequence data, PS3 performs additional computation to find a high-probability complete sequence for each partial sequence specified by an FSA. Because constrained beam search maintains a beam of $b$ sequences for each FSA state, this cost is given by $|S| \cdot b \cdot \gamma$, where $|S|$ is the number of FSA states, $b$ is the beam size parameter, and $\gamma$ is the computational cost of a single forward pass through an unrolled recurrent neural network (e.g., the cost of decoding a single sequence). Although the computational cost of training increases linearly with the number of FSA states, for any particular application FSA construction is a modeling choice and there are many existing FSA compression and state reduction methods available.

# 4   Application to image captioning

In this section, we describe how image captioning models can be trained on object annotations and image tags by interpreting these annotations as partially-specified caption sequences.

**Captioning model** For image captioning experiments we use the open source bottom-up and top-down attention captioning model [10], which we refer to as Up-Down. This model belongs to the class of 'encoder-decoder' neural architectures and recently achieved state of the art results on the COCO test server [6]. The input to the model is an image, $I$. The encoder part of the model consists of a Faster R-CNN object detector [47] based on the ResNet-101 CNN [48] that has been pre-trained on the Visual Genome dataset [49]. Following the methodology in Anderson et al. [10], the image $I$ is encoded as a set of image feature vectors, $V = \{\boldsymbol{v}_1, \ldots, \boldsymbol{v}_k\}, \boldsymbol{v}_i \in \mathbb{R}^D$, where each vector $\boldsymbol{v}_i$ is associated with an image bounding box. The decoder part of the model consists of a 2-layer Long Short-Term Memory (LSTM) network [50] combined with a soft visual attention mechanism [2]. At each timestep $t$ during decoding, the decoder takes as input an encoding of the previously generated word given by $W_e \Pi_t$, where $W_e \in \mathbb{R}^{M \times |\Sigma|}$ is a word embedding matrix for a vocabulary $\Sigma$ with embedding size $M$, and $\Pi_t$ is one-hot encoding of the input word at timestep $t$. The model outputs a conditional distribution over the next word output given by $p(y_t \mid y_{1:t-1}) = \mathrm{softmax}\,(W_p \boldsymbol{h}_t + \boldsymbol{b}_p)$, where $\boldsymbol{h}_t \in \mathbb{R}^N$ is the LSTM output and $W_p \in \mathbb{R}^{|\Sigma| \times N}$ and $\boldsymbol{b}_p \in \mathbb{R}^{|\Sigma|}$ are learned weights and biases. The decoder represents the distribution over complete output sequences using Equation 1.

**Finite state automaton construction** To train image captioning models on datasets of object detections and labeled images, we construct finite state automata as follows. At each training iteration we select three labels at random from the labels assigned to each image. Each of the three selected labels is mapped to a disjunctive set $D_i$ containing every word in the vocabulary $\Sigma$ that shares the same word stem. For example, the label `bike` maps to { `bike`, `bikes`, `biked`, `biking` }. This gives the captioning model the freedom to choose word forms. As the selected image labels may include redundant synonyms such as `bike` and `bicycle`, we only enforce that the generated caption mentions at least two of the three selected image labels. We therefore construct a finite state automaton that accepts strings that contain *at least one word* from *at least two* of the disjunctive sets. As illustrated in Figure 2(c), the resulting FSA contains eight states (although the four accepting states could be collapsed into one). In initial experiments we investigated several variations of this simple construction approach (e.g., randomly selecting two or four labels, or requiring more or fewer of the selected labels to be mentioned in the caption). These alternatives performed slightly worse than the approach described above. However, we leave a detailed investigation of more sophisticated methods for constructing finite state automata encoding observed partial information to future work.

**Out-of-vocabulary words** One practical consideration when training image captioning models on datasets of object detections and labeled images is the presence of out-of-vocabulary words. The constrained decoding in Step 1 can only produce fluent sentences if the model can leverage some side information about the out-of-vocabulary words. To address this problem, we take the same approach as Anderson et al. [13], adding pre-trained word embeddings to both the input and output layers of the decoder. Specifically, we initialize $W_e$ with pretrained word embeddings, and add an additional output layer such that $\boldsymbol{v}_t = \tanh\,(W_p \boldsymbol{h}_t + \boldsymbol{b}_p)$ and $p(y_t \mid y_{1:t-1}) = \mathrm{softmax}\,(W_e^T \boldsymbol{v}_t)$. For the word embeddings, we concatenate GloVe [37] and dependency-based [51] embeddings, as we find that the resulting combination of semantic and functional context improves the fluency of the constrained captions compared to using either embedding on its own.

**Implementation details** In all experiments we initialize the model by training on the available image-caption dataset following the cross-entropy loss training scheme in the Up-Down paper [10], and keeping pre-trained word embeddings fixed. When training on image labels, we use the online version of our proposed training algorithm, constructing each minibatch of 100 with an equal number of complete and partially-specified training examples. We use SGD with an initial learning rate of 0.001, decayed to zero over 5K iterations, with a lower learning rate for the pre-trained word embeddings. In beam search and constrained beam search decoding we use a beam size of 5. Training (after initialization) takes around 8 hours using two Titan X GPUs.

## 5 Experiments

### 5.1 COCO novel object captioning

**Dataset splits** To evaluate our proposed approach, we use the COCO 2014 captions dataset [52] containing 83K training images and 41K validation images, each labeled with five human-annotated

Table 1: Impact of training and decoding with image labels on COCO novel object captioning validation set scores. All experiments use the same finite state automaton construction. On out-of-domain images, imposing label constraints during training using PS3 always improves the model (row 3 vs. 1, 4 vs. 2, 6 vs. 5), and constrained beam search (CBS) decoding is no longer necessary (row 4 vs. 3). The model trained using PS3 and decoded with standard beam search (row 3) is closest to the performance of the model trained with the full set of image captions (row 7).

| | Training Captions | PS3 Labels | CBS Labels | Out-of-Domain Scores | | | | In-Domain Scores | | |
|---|---|---|---|---|---|---|---|---|---|---|
| | | | | SPICE | METEOR | CIDEr | F1 | SPICE | METEOR | CIDEr |
| 1 | ◖ | | | 14.4 | 22.1 | 69.5 | 0.0 | **19.9** | **26.5** | **108.6** |
| 2 | ◖ | | ▲ | 15.9 | 23.1 | 74.8 | 26.9 | 19.7 | 26.2 | 102.4 |
| 3 | ◖ | ● | | **18.3** | **25.5** | **94.3** | **63.4** | 18.9 | 25.9 | 101.2 |
| 4 | ◖ | ● | ▲ | 18.2 | 25.2 | 92.5 | 62.4 | 19.1 | 25.9 | 99.5 |
| 5 | ◖ | | ★ | 18.0 | 24.5 | 82.5 | 30.4 | 22.3 | 27.9 | 109.7 |
| 6 | ◖ | ● | ★ | 20.1 | 26.4 | 95.5 | 65.0 | 21.7 | 27.5 | 106.6 |
| 7 | ● | | | 20.1 | 27.0 | 111.5 | 69.0 | 20.0 | 26.7 | 109.5 |

● = full training set, ◖ = impoverished training set, ▲= constrained beam search (CBS) decoding with predicted labels, ★= CBS decoding with ground-truth labels

Table 2: Performance on the COCO novel object captioning test set. '+ CBS' indicates that a model was decoded using constrained beam search [13] to force the inclusion of image labels predicted by an external model. On standard caption metrics, our generic training algorithm (PS3) applied to the Up-Down [10] model outperforms all prior work.

| Model | CNN | Out-of-Domain Scores | | | | In-Domain Scores | | |
|---|---|---|---|---|---|---|---|---|
| | | SPICE | METEOR | CIDEr | F1 | SPICE | METEOR | CIDEr |
| DCC [21] | VGG-16 | 13.4 | 21.0 | 59.1 | 39.8 | 15.9 | 23.0 | 77.2 |
| NOC [22] | VGG-16 | - | 21.3 | - | 48.8 | - | - | - |
| C-LSTM [23] | VGG-16 | - | 23.0 | - | 55.7 | - | - | - |
| LRCN + CBS [13] | VGG-16 | 15.9 | 23.3 | 77.9 | 54.0 | 18.0 | 24.5 | 86.3 |
| LRCN + CBS [13] | Res-50 | 16.4 | 23.6 | 77.6 | 53.3 | 18.4 | 24.9 | 88.0 |
| NBT [24] | VGG-16 | 15.7 | 22.8 | 77.0 | 48.5 | 17.5 | 24.3 | 87.4 |
| NBT + CBS [24] | Res-101 | 17.4 | 24.1 | 86.0 | **70.3** | 18.0 | 25.0 | 92.1 |
| PS3 (ours) | Res-101 | **17.9** | **25.4** | **94.5** | 63.0 | **19.0** | **25.9** | **101.1** |

captions. We use the splits proposed by Hendricks et al. [21] for novel object captioning, in which all images with captions that mention one of eight selected objects (including synonyms and plural forms) are removed from the caption training set, which is reduced to 70K images. The original COCO validation set is split 50% for validation and 50% for testing. As such, models are required to caption images containing objects that are not present in the available image-caption training data. For analysis, we further divide the test and validation sets into their in-domain and out-of-domain components. Any test or validation image with a reference caption that mentions a held-out object is considered to be out-of-domain. The held-out objects classes selected by Hendricks et al. [21], are BOTTLE, BUS, COUCH, MICROWAVE, PIZZA, RACKET, SUITCASE, and ZEBRA.

**Image labels**  As with zero-shot learning [53], novel object captioning requires auxiliary information in order to successfully caption images containing novel objects. In the experimental procedure proposed by Hendricks et al. [21] and followed by others [13, 22, 23], this auxiliary information is provided in the form of image labels corresponding to the 471 most common adjective, verb and noun base word forms extracted from the held-out training captions. Because these labels are extracted from captions, there are no false positives, i.e., all of the image labels are salient to captioning. However, the task is still challenging as the labels are pooled across five captions per image, with the number of labels per image ranging from 1 to 27 with a mean of 12.

**Evaluation**  To evaluate caption quality, we use SPICE [54], CIDEr [55] and METEOR [56]. We also report the F1 metric for evaluating mentions of the held-out objects. The ground truth for an object mention is considered to be positive if the held-out object is mentioned in any reference

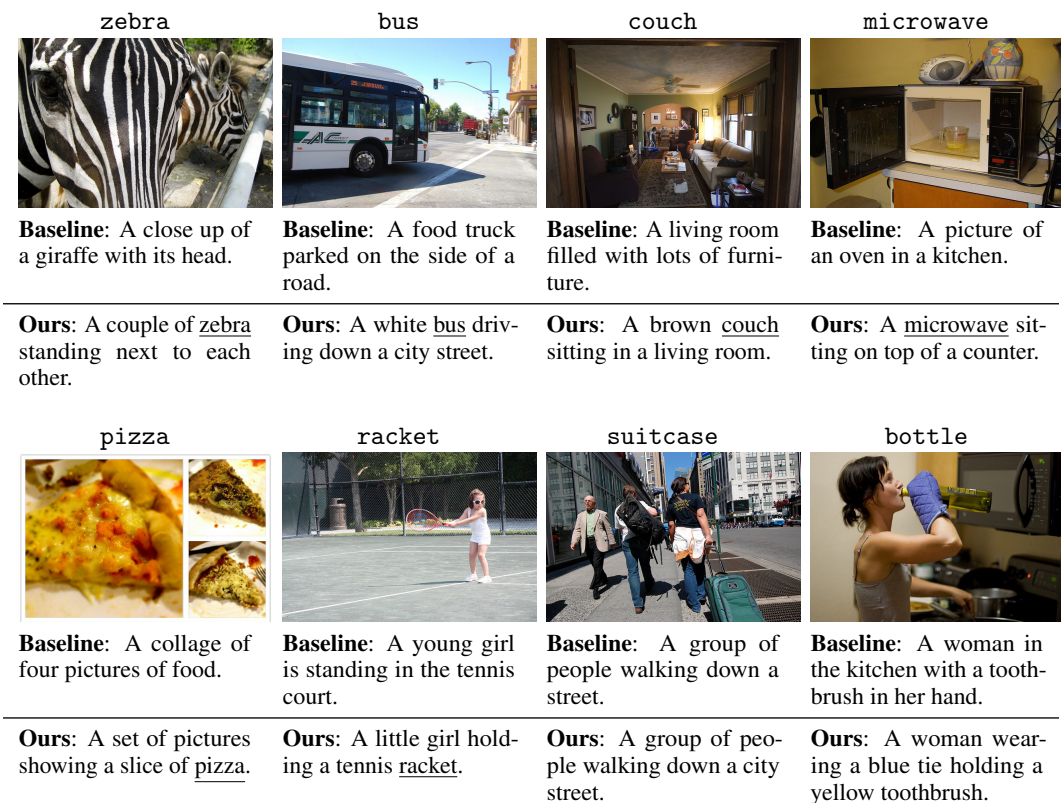

| zebra | bus | couch | microwave |
|---|---|---|---|
| **Baseline**: A close up of a giraffe with its head. | **Baseline**: A food truck parked on the side of a road. | **Baseline**: A living room filled with lots of furniture. | **Baseline**: A picture of an oven in a kitchen. |
| **Ours**: A couple of zebra standing next to each other. | **Ours**: A white bus driving down a city street. | **Ours**: A brown couch sitting in a living room. | **Ours**: A microwave sitting on top of a counter. |

| pizza | racket | suitcase | bottle |
|---|---|---|---|
| **Baseline**: A collage of four pictures of food. | **Baseline**: A young girl is standing in the tennis court. | **Baseline**: A group of people walking down a street. | **Baseline**: A woman in the kitchen with a toothbrush in her hand. |
| **Ours**: A set of pictures showing a slice of pizza. | **Ours**: A little girl holding a tennis racket. | **Ours**: A group of people walking down a city street. | **Ours**: A woman wearing a blue tie holding a yellow toothbrush. |

Figure 3: Examples of generated captions for images containing novel objects. The baseline Up-Down [10] captioning model performs poorly on images containing object classes not seen in the available image-caption training data (top). Incorporating image labels for these object classes into training using PS3 allows the same model to produce fluent captions for the novel objects (bottom). The last two examples may be considered to be failure cases (because the novel object classes, `suitcase` and `bottle`, are not mentioned).

captions. For consistency with previous work, out-of-domain scores are macro-averaged across the held-out classes, and CIDEr document frequency statistics are determined across the entire test set.

**Results** In Table 1 we show validation set results for the Up-Down model with various combinations of PS3 training and constrained beam search decoding (top panel), as well as performance upper bounds using ground-truth data (bottom panel). For constrained beam search decoding, image label predictions are generated by a linear mapping from the mean-pooled image feature $\frac{1}{k} \sum_{i=1}^{k} \boldsymbol{v}_i$ to image label scores which is trained on the entire training set. The results demonstrate that, on out-of-domain images, imposing the caption constraints during training using PS3 helps more than imposing the constraints during decoding. Furthermore, the model trained with PS3 has assimilated all the information available from the external image labeler, such that using constrained beam search during decoding provides no additional benefit (row 3 vs. row 4). Overall, the model trained on image labels with PS3 (row 3) is closer in performance to the model trained with all captions (row 7) than it is to the baseline model (row 1). Evaluating our model (row 3) on the test set, we achieve state of the art results on the COCO novel object captioning task, as illustrated in Table 2. In Figure 3 we provide examples of generated captions, including failure cases. In Figure 4 we visualize attention in the model (suggesting that image label supervision can successfully train a visual attention mechanism to localize new objects).

## 5.2 Preliminary experiments on Open Images

Our primary motivation in this work is to extend the visual vocabulary of existing captioning models by making large object detection datasets available for training. Therefore, as a proof of concept

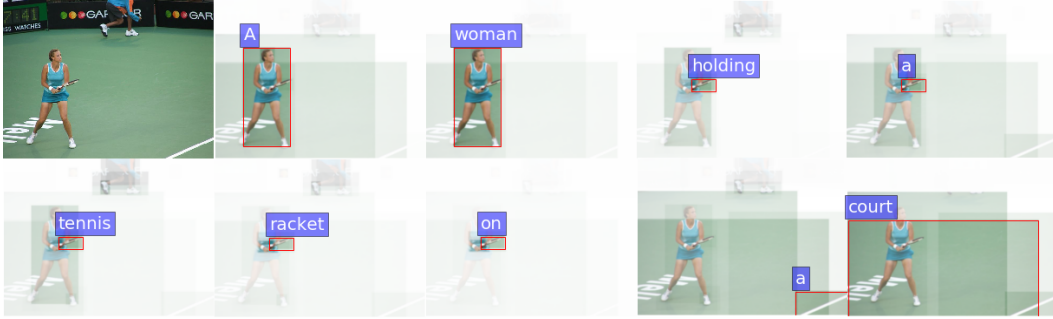

A woman holding a tennis racket on a court.

Figure 4: To further explore the impact of training using PS3, we visualize attention in the Up-Down [10] model. As shown in this example, using only image label supervision (i.e., without caption supervision) the model still learns to ground novel object classes (such as `racket`) in the image.

| tiger | monkey | rhino | rabbit |
|---|---|---|---|

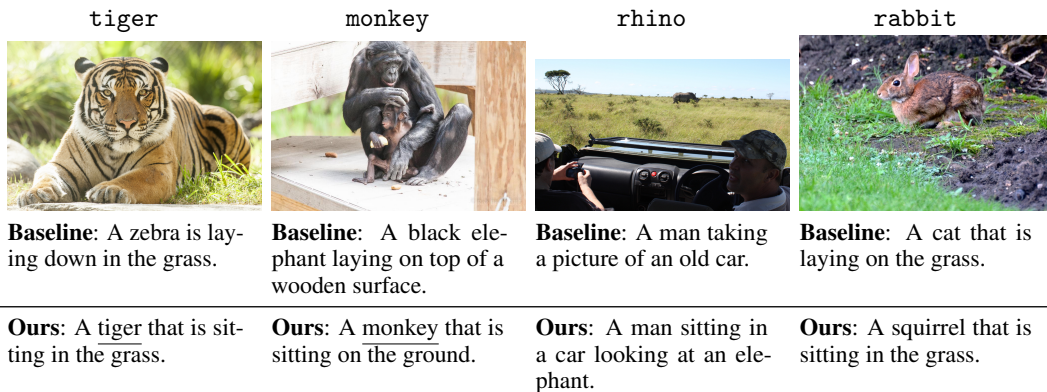

| **Baseline**: A zebra is lay-ing down in the grass. | **Baseline**: A black ele-phant laying on top of a wooden surface. | **Baseline**: A man taking a picture of an old car. | **Baseline**: A cat that is laying on the grass. |
|---|---|---|---|
| **Ours**: A tiger that is sit-ting in the grass. | **Ours**: A monkey that is sitting on the ground. | **Ours**: A man sitting in a car looking at an ele-phant. | **Ours**: A squirrel that is sitting in the grass. |

Figure 5: Preliminary experiments on Open Images. As expected, the baseline Up-Down [10] model trained on COCO performs poorly on novel object classes from the Open Images dataset (top). Incorporating image labels from 25 selected classes using PS3 leads to qualitative improvements (bottom). The last two examples are failure cases (but no worse than the baseline).

we train a captioning model simultaneously on COCO Captions [6] and object annotation labels for 25 additional animal classes from the Open Images V4 dataset [14]. In Figure 5 we provide some examples of the generated captions. We also evaluate the jointly trained model on the COCO 'Karpathy' val split [27], achieving SPICE, METEOR and CIDEr scores of 18.8, 25.7 and 103.5, respectively, versus 20.1, 26.9 and 112.3 for the model trained exclusively on COCO.

## 6 Conclusion

We propose a novel algorithm for training sequence models on partially-specified data represented by finite state automata. Applying this approach to image captioning, we demonstrate that a generic image captioning model can learn new visual concepts from labeled images, achieving state of the art results on the COCO novel object captioning splits. We further show that we can train the model to describe new visual concepts from the Open Images dataset while maintaining competitive COCO evaluation scores. Future work could investigate training captioning models on finite state automata constructed from scene graph and visual relationship annotations, which are also available at large scale [14, 49].

**Acknowledgments**

This research was supported by a Google award through the Natural Language Understanding Focused Program, CRP 8201800363 from Data61/CSIRO, and under the Australian Research Council's Discovery Projects funding scheme (project number DP160102156). We also thank the anonymous reviewers for their valuable comments that helped to improve the paper.

## Footnotes

*Now at Georgia Tech (peter.anderson@gatech.edu)

[2]www.panderson.me/constrained-beam-search

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
