[Supplementary Material]

# Supplementary Material for Partially-Supervised Image Captioning

**Peter Anderson**
Macquarie University*
Sydney, Australia
p.anderson@mq.edu.au

**Stephen Gould**
Australian National University
Canberra, Australia
stephen.gould@anu.edu.au

**Mark Johnson**
Macquarie University
Sydney, Australia
mark.johnson@mq.edu.au

As supplementary material we provide additional caption examples for COCO novel object captioning in Figure 1, and for captions trained with Open Images in Figure 2. Further analysis of the impact of adding pre-trained word embeddings to the base model is included in Table 1.

Table 1: Analysis of the impact of adding fixed word embeddings (GloVe [1], dependency embeddings [2] or both) to the Up-Down [3] captioning model. T$x$of$y$ indicates the model was decoded using constrained beam search [4] requiring the inclusion of at least $x$ of the top $y$ concepts randomly selected from the ground-truth image labels. Adding fixed embeddings has a slightly negative impact on the model when decoding without constraints (top panel). However, concatenating both embeddings (capturing both semantic and functional information) helps to preserve fluency during constrained decoding (bottom two panels).

| Model | Out-of-Domain Val Scores | | | | In-Domain Val Scores | | |
|---|---|---|---|---|---|---|---|
| | SPICE | METEOR | CIDEr | F1 | SPICE | METEOR | CIDEr |
| Up-Down | 14.4 | 22.1 | 69.5 | 0.0 | 19.9 | 26.5 | 108.6 |
| Up-Down-GloVe | 14.0 | 21.6 | 66.4 | 0.0 | 19.5 | 26.2 | 104.1 |
| Up-Down-Dep | 14.3 | 21.9 | 67.9 | 0.0 | 19.4 | 26.0 | 105.0 |
| Up-Down-Both | 14.0 | 21.8 | 66.7 | 0.0 | 19.5 | 26.1 | 104.0 |
| Up-Down-GloVe + T2of3 | 18.0 | 24.4 | 80.2 | 28.3 | 22.2 | **27.9** | 109.0 |
| Up-Down-Dep + T2of3 | 17.8 | 24.4 | 79.5 | 23.8 | 21.8 | 27.5 | 107.3 |
| Up-Down-Both + T2of3 | **18.3** | **24.9** | **84.1** | **31.3** | **22.3** | 27.8 | **109.4** |
| Up-Down-GloVe + T3of3 | 19.0 | 24.6 | 80.1 | 45.2 | **23.0** | 27.4 | 101.4 |
| Up-Down-Dep + T3of3 | 19.0 | 24.5 | 79.0 | 42.2 | 22.3 | 26.9 | 98.4 |
| Up-Down-Both + T3of3 | **19.6** | **25.1** | **82.2** | **45.8** | **23.0** | 27.5 | **102.2** |

**zebra**     **bus**     **couch**     **microwave**

**Baseline**: A group of giraffes standing next to each other.

**Ours**: A group of zebra standing next to each other.

**Baseline**: A group of people standing in front of a building.

**Ours**: A group of people standing next to a bus.

**Baseline**: A living room filled with furniture and a chair.

**Ours**: A white couch sitting in a living room.

**Baseline**: A kitchen with wood cabinets and wooden appliances.

**Ours**: A kitchen with a stainless steel refrigerator.

**Baseline**: A black and white photo of a giraffe eating grass.

**Ours**: A zebra standing in a field eating grass.

**Baseline**: A yellow truck with graffiti on the road.

**Ours**: A yellow bus driving down a city street.

**Baseline**: A brown and white dog laying on a bed.

**Ours**: A brown and white dog sitting on a couch.

**Baseline**: A picture of a kitchen with an oven.

**Ours**: A microwave oven sitting on display.

**pizza**     **racket**     **suitcase**     **bottle**

**Baseline**: A man and a woman eating food at a table.

**Ours**: A woman sitting at a table eating pizza.

**Baseline**: A man standing in front of a white fence.

**Ours**: A man holding a tennis racket on a court.

**Baseline**: A man and a woman standing next to a car.

**Ours**: A woman standing next to a man holding a suitcase.

**Baseline**: A person sitting on top of a laptop computer.

**Ours**: A person sitting next to a computer keyboard.

**Baseline**: A piece of food is on a plate.

**Ours**: A piece of pizza sitting on top of a white plate.

**Baseline**: A young girl playing a game of tennis.

**Ours**: A girl hitting a tennis ball on a court.

**Baseline**: A cat laying on top of a bag.

**Ours**: A cat sitting on top of a suitcase.

**Baseline**: Two glasses of wine are sitting on a table.

**Ours**: A glass of wine sitting on top of a table.

Figure 1: Further examples of captions generated by the Up-Down captioning model (top) and the same model trained with additional image labels using PS3 (bottom). All images shown contain held-out objects.

| koala | goat | deer | monkey |
|---|---|---|---|

**Baseline**: A tree that is standing next to a branch.

**Ours**: A monkey that is sitting in a tree.

**Baseline**: A brown and white dog laying in a pen.

**Ours**: A goat that is sitting in the grass.

**Baseline**: A herd of animals that are standing in the grass.

**Ours**: A deer that is sitting in the grass.

**Baseline**: Two brown bears are playing in the water.

**Ours**: A monkey that is sitting in the water.

| squirrel | lion | rabbit | lion |
|---|---|---|---|

**Baseline**: A cat sitting on top of a tree branch.

**Ours**: A squirrel that is sitting on a tree.

**Baseline**: A statue of a bear in front of a building.

**Ours**: A statue of a lion that is sitting on a tree.

**Baseline**: A close up of a black and white cat.

**Ours**: A close up of a black cat sitting on the floor.

**Baseline**: A statue of an elephant sitting on a sidewalk.

**Ours**: A statue of a lion sitting on a cobblestone sidewalk.

Figure 2: Further examples of captions generated by the Up-Down captioning model trained on COCO (top) and the same model trained with COCO and image labels from an additional 25 Open Images animal classes using PS3 (bottom). Several examples are failure cases (but no worse than the baseline).

## Footnotes

*Now at Georgia Tech (peter.anderson@gatech.edu)