[Reviews · NeurIPS 2018]

Reviewer 1



This paper proposes to teach image captioning models new visual concepts with partial supervision, i.e. image level labels. Based on the image labels, a complete caption is first estimated using constrained decoding. Then the image caption model is trained using both supervised data and the estimated image-sentence data. The proposed method achieves the state-of-the-art result on novel object captioning task. Strengths: 1. It is novel to apply constrained beam search [19] in the training process. 2. The experimental results are good. Weaknesses: 1. The writing is not clear. The descriptions of the techniqual part can not be easily followed by the reviewer, which make it very hard to reimplement the techniques. 2. An incomplete sequence is represented by a finite state automaton. In this paper, only a two out of three finite state automation is used. Is it possible/computational feasible to use more complicated finite state automaton to select more image labels? As there are 12 image labels per image on average, only selecting two labels seems insufficient. 3. The authors describe an online version of the algorithm because it is impractical to train multiple iterations/epochs with large models and datasets. Is it true that the proposed method requires much more computation than other methods? Please compare the computational complexity with other methods. 4. How many estimated complete captions could be obtained for one image? Is it possible to generate over C(12, 3) * C(3, 2) = 660 different captions for one image?

Reviewer 2



The authors tackle the problem of image captioning in the setting where captions may not be available for `novel' categories. This setting is well motivated by the authors and is a practical one - it is easier to scale image-level annotations than it is to get image captions. They present an approach that uses some paired image captions and visual labels of novel categories to caption novel categories. The visual labels of novel categories are combined using an automata to only accept captions that contain some mention of the new classes. The method is evaluated on the novel class split proposed by Hendricks et al. The authors show numerical improvement over baselines. Strengths - The paper is clearly written and it was fun to read. I like the way the authors have motivated the problem, their setup and solution. The use of an automata is a nice idea. - The authors have done a good job of evaluating against recent state-of-the-art baselines, including papers published at CVPR 2018 (NBT). - The qualitative examples in Figure 2 and 4 are useful to see what this method can achieve and where it fails. - Table 1 does a good job of highlighting the upper bound of this method, and showing how close one can get to it using the partial supervision approach. Weaknesses - L198-205: It is not clear how exactly the FSA is constructed. Do you select 3 labels out of all the visual labels (novel ones) for the image? Is this selection ever changed? How are synonyms determined? How is the FSA constructed to handle synonyms - do you add synonyms to all the disjunctions and create separate sets? The paper spends little time explaining details of it's major idea/contribution. This makes reproducibility hard, although the authors do say they will release code (L83). - A minor observation from Table 1: the in-domain score is highest when not using any out-of-domain data (row 1). Does this indicate that model capacity is limited because using out-of-domain data reduces in-domain (especially for CIDEr) performance. - A missing comparison from Table 2 is using LRCN+CBS with a ResNet-101 CNN. Moving from VGG16 to ResNet101 gives significant improvement for the NBT baseline, and I suspect it should do the same for LRCN. A comprehensive comparison would benefit the authors and future research. After rebuttal ------------------ I do not agree with the authors' assertion that ResNet-50 results for LRCN "should be similar to ResNet-101". I strongly suggest running the experiment instead of suggesting similarity. I will keep my rating unchanged.

Reviewer 3



The paper works on the problem of captioning novel object in images without paired image-caption training data, assuming object labels but no captions are available for the objects. The paper proposes to use constrained beam search [19] as a subroutine at training time to generate a caption with includes the object labels for an image, relying on a finite state machine to constrain the beam search with multiple labels. This allows to generate new training data, and an EM algorithm is used to generate training data and learn a better captioning model The approach is more general in the sense that it could not just handle object labels but other forms of incomplete captions, e.g. phrases, however, this is not evaluated in the paper. Strength: - Novel and interesting direction to approach novel image object captioning - Evaluation which includes ablations and comparison to state-of-the art, including very recent work from CVPR 2018 [23]. - Outperforms all prior work (one small exception is F1 Measure, where [23]+CBS is higher) Weaknesses: 1. While authors discuss [19], it is not obvious in section 3.1. (and line 57) that [19] uses same Fine state automaton (or is there any difference?). 2. It would have been interesting to compare to the same out of coco test as in [20,21] with F1 score for better comparability. 3. It would be interesting to see (qualitatively) what kind of captions the CBS + fine state machine gives. I guess this is quantitatively shown in row 6, Table 1? Why is the F1 score in this case not significantly higher although gt labels are used? 4. Figure 1(b): It would be great if the authors would clarify the use of the example in Figure 1(b), in general and specifically if this is also used for captioning. As well as which sequences this automaton would allow, anything which starts with “the” but does not contain “score”? a. More state-machine examples would have been interesting in supplemental, also jointly with image + gt labels examples + generated caption. 5. Minor: line 40: “This reflects the observation that, in general, object detection datasets are easier to scale”. I don’t agree that there is much evidence for this. I think it is rather an artifact on what the community focuses on. The authors might want to consider reducing the strength of this statement. However, I do strongly agree that image captioning should be able to exploit detection data as done in this work. Conclusion: The paper presents an interesting approach for novel object image captioning with strong experimental validation which I advocate to accept. ---- After going through the author response and reviews: + Overall the authors addressed mine and other reviewers questions and promised to add further comparisons in the final. - unfortunately, the authors did not provide the additional results already in the author response. Both for mine and the concern of not comparable networks for LRCN+CBS, brought up by R2, which I agree to. In conclusion I recommend acceptance, in expectation that the authors will add the promised results to the final version. I also suggest to include comparable results for LRCN+CBS or a fair ablation of the author's model with the same image network.